# Fairness-guided Few-shot Prompting for Large Language Models

**Huan Ma[1,2], Changqing Zhang[1]\*, Yatao Bian[2], Lemao Liu[2], Zhirui Zhang[2],**
**Peilin Zhao[2], Shu Zhang[2], Huazhu Fu[3], Qinghua Hu[1], Bingzhe Wu[2]\***
[1] College of Intelligence and Computing, Tianjin University, Tianjin, China
[2] AI Lab, Tencent, Shenzhen, China
[3] Institute of High Performance Computing (IHPC),
Agency for Science, Technology and Research (A*STAR), Singapore
[1] zhanchangqing@tju.edu.cn; [2] bingzhewu@tencent.com

## Abstract

Large language models have demonstrated surprising ability to perform in-context learning, i.e., these models can be directly applied to solve numerous downstream tasks by conditioning on a prompt constructed by a few input-output examples. However, prior research has shown that in-context learning can suffer from high instability due to variations in training examples, example order, and prompt formats. Therefore, the construction of an appropriate prompt is essential for improving the performance of in-context learning. In this paper, we revisit this problem from the view of predictive bias. Specifically, we introduce a metric to evaluate the predictive bias of a fixed prompt against labels or a given attributes. Then we empirically show that prompts with higher bias always lead to unsatisfactory predictive quality. Based on this observation, we propose a novel search strategy based on the greedy search to identify the near-optimal prompt for improving the performance of in-context learning. We perform comprehensive experiments with state-of-the-art mainstream models such as GPT-3 on various downstream tasks. Our results indicate that our method can enhance the model's in-context learning performance in an effective and interpretable manner. Code is available at: https://github.com/MaHuanAAA.

## 1   Introduction

Large language models (LLMs), such as GPT-3 [1] and BLOOM [2], have demonstrated remarkable ability in performing in-context learning (ICL) on downstream tasks. ICL refers to the process of conditioning an LLM to solve various downstream tasks using prompts constructed from a few demonstration input-output pairs [3] (i.e., few-shot prompting). Despite its impressive performance, prior research has shown that ICL suffers from high instability due to variations in the choice of in-context demonstrations, demonstration order, and prompt formats [4, 5]. Therefore, constructing an appropriate prompt has been identified as a critical factor for improving the performance of ICL [6].

Previous research studies this problem typically from two directions: (1) prompt tuning in the embedding space [7, 8, 9, 10, 11], (2) prompt searching in the text space [4, 12, 13, 14, 15, 16]. The key idea of prompt tuning is to inject task-specific embedding into hidden layers and then tune these embeddings using gradient-based optimization [8, 15]. However, these methods require to modify the original inference process of the model, which is impractical for the case of black-box LM services

---

*Corresponding author

37th Conference on Neural Information Processing Systems (NeurIPS 2023).

such as GPT3 and ChatGPT [17]. Furthermore, prompt tuning introduces additional computational and storage costs, which is typically expensive for LLM. A more feasible and efficient way is to optimize prompting via searching approximate demonstration samples and ordering in the original text space [4, 15]. Bunch of works are presented to constructs prompts from either "global" or "local" views. On the one hand, global-view based methods typically optimize the different elements of the prompt as a whole, with the aim of achieving superior performance. For example, one approach, as described in [14], constructs a search procedure that leverages the overall diversity of demonstrations. Another approach [4] attempts to optimize the ordering of the entire set of demonstrations to achieve better performance. In contrast to the global view, local-view based methods optimize each individual demonstration by designing different heuristic selection criteria such as prior work KATE [15]. These methods have achieved impressive improvements on a wide range of tasks. However, most of them still suffer from the following limitations: (1) Most of current research mainly focuses on searching prompts along a single dimension, such as example selection or order. However, the overall influence of various dimensions on the performance remains unclear. (2) These methods are typically based on heuristic criteria, and there is a gap between them and actual performance. A unified view that explains how these methods work is needed. (3) More importantly, existing methods optimize prompts globally or locally, which may lead to suboptimal performance.

In this paper, we revisit this problem from the perspective of *predictive bias*. We find a key insight that the quality of a given prompt depends on its inherent bias. Based on this insight, we propose a surrogate metric based on predictive bias for evaluating the quality of prompts. This metric allows us to evaluate a prompt in a single forward process without an additional development set. Specifically, we apply a given prompt to a "content-free" input and expect the model output an uniform predictive distribution (a content-free input contains no useful information). Therefore, we employ the uniformity of the predictive distribution to characterize the bias of a give prompt. This shares a similar idea to the prior work which uses this metric to calibrate the model output [18]. In contrast to this work which mainly focus on using this metric for calibration when the prompt is fixed, we further explore its usage in automatically searching an approximate prompt. Moreover, through extensive experiments, we empirically validate the correlation between the inherent bias of a given prompt and its quality measured by the average task performance on a given test set (see Fig. 2).

Moreover, this bias-based metric allows us to build prompting optimization techniques in a "local-to-global" manner. We present two novel strategies for efficiently searching high-quality prompts in a bias-guided way: (1) T-fair-Prompting (2) G-fair-Prompting. We focus on a general setting where a labeled set with size $N$ is given. The goal of our strategies is to perform combinatorial optimization over this set to find near-optimal prompts (i.e., select demonstrations and their orders). Specifically, T-fair-Prompting uses an intuitive way that first computes the bias of each single demonstration (i.e., one-shot prompting) and then select the top-k fair demonstrations to form the final prompts. This strategy can be efficiently done with a complexity of $O(N)$. Note that T-fair-Prompting is based on the assumption that the optimal prompt is usually constructed from demonstrations with the smallest individual bias. However, this may not hold true in real situations and often leads to sub-optimal solutions. Therefore, we further introduce G-fair-Prompting to improve the search quality. G-fair-Prompting follows the normal procedure of the greedy search which finds the optimal solution by making locally optimal choices at each step. At each step of the algorithm, the selected demonstration is the one which makes the updated prompts achieves the best fairness score.This strategy trades off the quality of the search with the worst-case time complexity. By accepting a higher worst-case time complexity of $O(N^2)$, the search quality is significantly improved. Note that G-fair-Prompting works from a local to global perspective, wherein bias of individual samples are considered in the early stages while the later stage focus on the reduction of global predictive bias.

To evaluate the effectiveness of our strategies, we conduct extensive experiments with current mainstream models, such as GPT-3 [1], on various downstream tasks. Our results indicate that our method can significantly enhance the model's in-context learning performance in an effective and interpretable manner. The overall contribution is summarized as follows:

- We introduce to use the predictive bias to assess the quality of a given prompt in an efficient and development set independent way and the empirical effectiveness of this metric is comprehensively validated.

- Based on the above idea, we propose two efficient and effective strategies, namely, T-fair-Prompting and G-fair-Prompting to optimize the prompts.

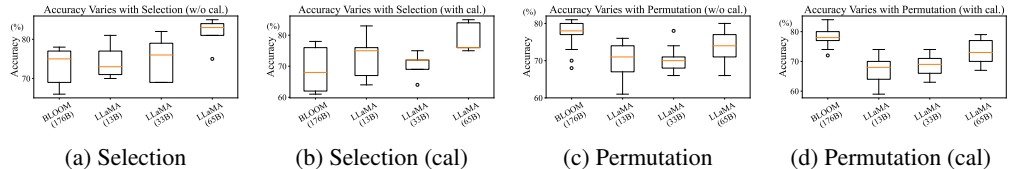

Figure 1: ICL suffers from high instability due to high variations in demonstrations selection and order, even when post calibration is performed.

- The effectiveness of these two strategies are validated on various LLMs ranging from GPT-series models to LLaMA family [19] released by Meta recently. Consistent relative improvements of over $10\%$ have been observed over different downstream tasks in contrast to SOTA methods.

**Relation to Calibration-before-use:** Our paper shares a similar metric with cal-before-use [18] to asses the predictive bias of a given prompt. However, the prior approach aims to use this metric to calibrate the output, which can be still easily affected by the quality of the used prompt (more results can be found in Table 2). In contrast, our research aims to find a near-optimal prompt on the original space to improve the model's performance, without requiring any post-adjustment to the output of the model. Moreover, we have firstly empirically validated the connection between predictive bias and the final task performance as shown in Fig. 2, which has not been studied in [18]. Through experiments, we have discovered that, even without calibration, the prompt selected by our method can outperform a randomly selected prompt with calibration.

## 2 Related Work

**In-context Learning** Previous research, as cited in [1, 20], has demonstrated that Large Language Models can complete tasks with zero- or few-shot learning using in-context learning. LLMs perform well with an appropriate prompt. However, recent works [4, 18] has shown that the performance of LLMs is affected by the prompt used. Therefore, determining the optimal prompt is a crucial and fundamental research area.

**Original space searching** A more intuitive approach for determining the best prompt is to search in the original space by selecting or reordering the prompt sentences entered by users. The searching can be concluded in two perspective. • **Global view**: A naive strategy is to *enumerate* all candidates to find the prompt that can achieve the best performance on validation set, but this strategy is computationally expensive since its complexity is $\sum_{k=1}^{n} C_n^k k!$ considering to demonstration selection and order permutation, where $k$ represents the number of demonstrations selected, and $C$ signifies the combinatorial function. Zhang et al. [12] find that errors frequently fall into the same cluster, where each cluster contains similar questions, so they proposed a *diversity-guided* searching strategy to select diverse demonstrations. In addition to demonstrations selection, Lu et al. [4] have identified the impact of the prompt *order* on the results. They found the best sequence which yields the most diverse prediction results on the probing set by generating a probing set through LLMs. However, this method is also computationally expensive, and it may be difficult to ensure that the generated probing set is sufficiently balanced. • **Local view**: Previous studies [13] show that reducing the model's *uncertainty* helps improve the model's performance, and [14] propose Active Prompting to select demonstrations according to the uncertainty of LLMs. KATE [15] selects the prompt based on the *distance* amongst embeddings, with the goal of selecting the closest example. However, this method ignores the influence of the order of the examples and requires access to sentence embeddings. [16] demonstrate that LLMs can be easily distracted by irrelevant context, accordingly they identify several approaches for *filtering* out irrelevant information in context.

In the realm of original space searching, most of the current methods tend to focus solely on the influence of a singular factor (highlighted above) on performance, utilizing heuristic metrics to select context demonstrations that perform well according to this criterion. While these investigations certainly bring benefits to the community, they lack a comprehensive consideration of both local and

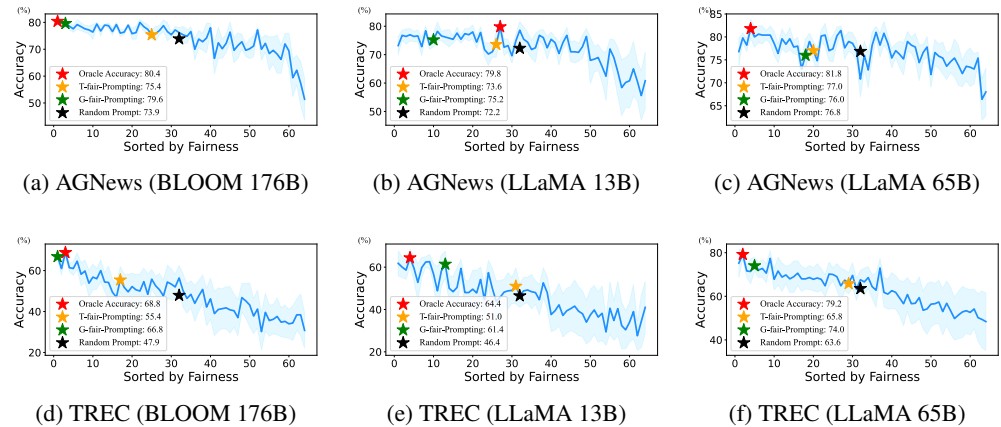

Figure 2: Accuracy is highly consistency with fairness and greedy search can find a good prompt, where "Random" and "Oracle" indicate the average accuracy of all prompts and the upper-bound performance according to fairness.

global perspectives. The method proposed offers a metric to select context demonstrations from the perspective of predictive bias, which naturally facilitates a transition from local view to global view.

## 3    Revisiting the Sensitivity across Demonstrations

In this section, we will clarify the notations and the templates used in this paper. Then, we will demonstrate some brief empirical results to show how different demonstration construction factors (e.g., example selection and order) affect performance. We further introduce the definition of predictive bias/fairness of a given prompt and show its connection to the predictive performance on different downstream tasks.

### 3.1    Notations

We consider a training set consisting of $N$ samples $S = \{(x_i, y_i)\}_{i=1}^N$, where $x_i$ is the sentence and $y_i \in \mathcal{Y}$ is the label of the $i^{th}$ training sample, and $\mathcal{Y}$ is the space of all labels for the task. We use a template $\Gamma(\cdot)$ to transform these sentences and labels into natural language space (i.e., prompt construction). Take an instance from the AGNews dataset [21] for example, we have $x_i$ = *"Cubans Risking Life for Lure of America."*, $y_i$ = *"World"*, and $\Gamma(x_i, y_i)$ is *"Article: Cubans Risking Life for Lure of America. Answer: World"*. We concatenate these demonstrations to form a prompt $\rho$, which by default is $\rho = \Gamma(x_1, y_1) \oplus \cdots \oplus \Gamma(x_n, y_n)$, where $\oplus$ indicates sentences combination option. At test time, we append the prompt $\rho$ with $\tau$ = *"Article: <test sentence>. Answer: "* and feed it to a large language model $\mathcal{M}$. The predicted class is given by:

$$\hat{y} = \arg\max_{y \in \mathcal{Y}} \hat{p}(y|\rho \oplus \tau), \quad \hat{p}(y|\rho \oplus \tau) = \frac{\mathcal{M}(y|\rho \oplus \tau)}{\sum_{y \in \mathcal{Y}} \mathcal{M}(y|\rho \oplus \tau)}, \tag{1}$$

where $\mathcal{M}(y|\rho \oplus \tau)$ indicates the probability predicted by LLM, and the probability is normalized to fit the task. We denote the predictive distribution by $\hat{P}(x) := \{\hat{p}(y|\rho \oplus \tau)|y \in \mathcal{Y}\}$. In this paper, we focus on evaluating the instability caused by demonstrations, and we fix the prompt template following prior work [18].

### 3.2    Stability of Few-shot Prompting

As demonstrated by prior research, the few-shot prompting technique is highly susceptible to a variety of factors, including the selection and order of demonstrations [4, 18]. In this study, we delve deeper into the stability of few-shot prompting, specifically focusing on the recently released LLaMA family

by Meta [19]. Additionally, we evaluate the stability of LLaMA models calibrated using the current state-of-the-art method [12, 15].

To elucidate the impact of demonstration selection, we select four demonstrations for each different seed and randomly sample an order for each combination. Subsequently, we present the performance on AGNews in the form of a boxplot, which displays the data distribution based on a five-number summary (minimum, first quartile [Q1], median, third quartile [Q3], and maximum). As shown in Fig.1(a)(b), the accuracy demonstrates significant variability across various demonstrations. The detailed settings please refer to Appendix A.7.

To investigate the influence of permutations, we examine all possible permutations of four fixed demonstrations, resulting in 4! distinct candidates. Fig.1(c)(d) also reveals a high degree of variance. While post-calibration contributes to mitigating instability, it is essential to note that the model remains sensitive even after post-calibration. This finding underscores the importance of meticulous demonstration selection. In subsequent experiments, we discover that our approach can be employed to further enhance the performance of the calibrated model.

### 3.3 Predictive Bias of ICL

As demonstrated in the preceding discussion, the performance of ICL is significantly impacted by various factors such as demonstration, permutation, and selection (refer to Appendix A.4 for additional information). Consequently, devising an efficient method for constructing an appropriate prompt with near-optimal performance is a crucial step in deploying LLMs for diverse downstream tasks. As outlined in the introduction, numerous studies aim to optimize prompts in ICL. This paper further investigates this issue through the lens of predictive bias, which refers to the discrepancy between targeted classes. [2]

To achieve this, we initially introduce an efficient technique to assess the inherent predictive bias of a given prompt, drawing inspiration from previous work [18]. We construct a training set-independent metric to measure predictive bias as follows: first, we merge the provided prompt with "semantic-free" test sample information (e.g., "[N/A]", denoted by $\eta$) and obtain the LLM's predictive distribution for this sample. Ideally, the predictive distribution should closely resemble a uniform distribution, as the test sample lacks semantic information. In this paper, we employ entropy as a measure of predictive bias, defined as:

$$\text{fair}(\rho) = -\sum_{y \in \mathcal{Y}} p(y|\rho \oplus \eta) \log p(y|\rho \oplus \eta) \tag{2}$$

Previous studies have utilized this metric to calibrate the model's output. In this paper, we conduct a comprehensive examination of the relationship between predictive bias and overall performance. Specifically, in a scenario with four training samples (due to the time-consuming nature of enumerating all prompt cases for a larger number), we enumerate all possible combinations and permutations of demonstrations for various datasets and LLMs. Subsequently, we arrange all candidates in descending order based on fairness, where an "index 0" denotes the prompt with the highest fairness. We perform experiments using five different seeds, resulting in training sets comprising distinct demonstrations while maintaining the test samples with seed 0. Fig. 2 displays the results for different models, revealing a strong correlation between the model's performance and fairness score (i.e., fairer prompts yield better performance). The red star, referred to as the "Oracle" represents the optimal average performance, which consistently correlates with higher fairness. This observation prompts us to enhance the ICL performance by identifying the fairest prompt.

Nevertheless, discovering the fairest demonstration combination proves to be a formidable challenge, given the existence of $\sum_{k=1}^{N} C_N^k k!$ distinct candidates. As the size of the training set increases, this task becomes intractable. In order to tackle this problem, we propose two efficient strategies for approximating the most suitable demonstrations in the subsequent section.

---

[2]This notion differs slightly from the concept of social bias, which concentrates on specific feature attributes rather than labels. Our approach can be naturally extended to mitigate social bias in various settings.

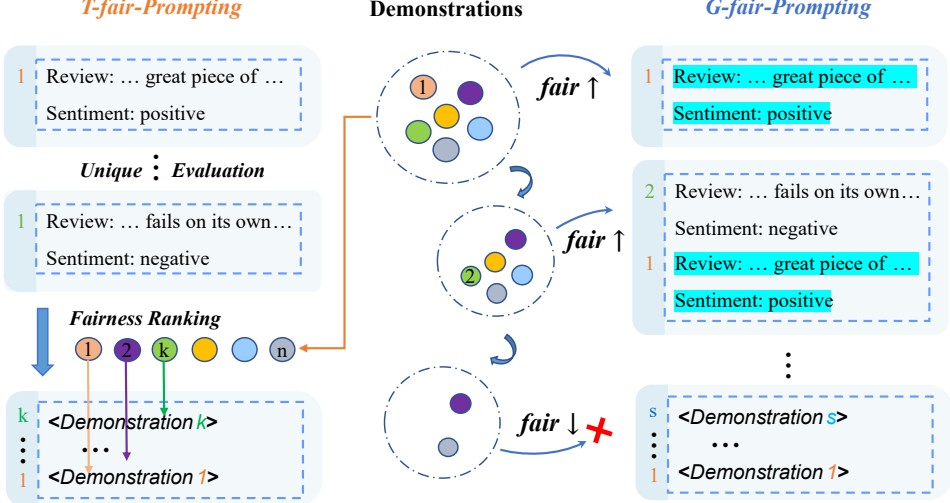

Figure 3: Overview of Most-fair Prompting.

## 4 Fairest Prompt Search

Drawing upon the aforementioned observations, we propose two strategies aimed at identifying the most fair prompt, which have been empirically demonstrated to achieve superior performance. Let us consider a training set $S$ comprising $N$ samples; the goal of these search strategies is to select a subset of samples from the training set and construct the context in a specific order so as to optimize the fairness criterion in Eq. 2.

In an ideal scenario, we would consider the factors of demonstration selection and order permutation by examining $\sum_{k=1}^{N} C_N^k k!$ distinct candidates, which enumerates all possible situations. However, evaluating every candidate is infeasible, as demonstrated when $N = 8$, yielding over $10^6$ candidates. In this paper, we introduce two search strategies to reduce computational cost: T-fair-Prompting and G-fair-Prompting. The T-fair-Prompting strategy decreases complexity from $\Theta(\sum_{k=1}^{N} C_N^k k!)$ to $\Theta(N)$, but its performance hinges on the selection of $k$ and may be unstable when an unsuitable value of $k$ is chosen. As a result, we propose an additional greedy search strategy, termed G-fair-Prompting, which lowers complexity to $O(N^2)$ and offers a superior approximation of the oracle solution. Fig. 9 visualizes the computational costs over different training set size.

### 4.1 T-fair-Prompting

The central idea of T-fair-Prompting (top-k) is founded on the heuristic understanding that the fairest prompt usually consists of demonstration samples with reduced individual biases. Consequently, T-fair-Prompting constructs the prompt through a two-stage process. Initially, the prediction bias is assessed when the prompt is formulated using individual demonstrations. Subsequently, the top-$k$ fairest demonstrations are chosen and employed to prompt the LLM. It is important to note that fairer demonstrations are likely to be situated towards the end of the sequence, as the generation is more influenced by proximate demonstrations, in accordance with prior research [18]. A comprehensive description of the process is presented in Algorithm 1, while a visual representation can be found in Fig. 3. Specifically, when $k$ is equivalent to the size of the training set, the method degrade to a search for the optimal order of demonstrations. Nevertheless, T-fair-Prompting is heavily reliant on the chosen value of $k$. More crucially, T-fair-Prompting addresses this issue through a purely local perspective, thereby neglecting considerations from a global standpoint, which typically results in sub-optimal outcomes. As a result, we subsequently introduce the G-fair-Prompting method, which operates in a local-to-global fashion, as described below.

## 4.2 G-fair-Prompting

The G-fair-Prompting (greedy search) algorithm adheres to the standard procedure of greedy search, which seeks the optimal solution by making locally optimal choices at each stage. In each step of the algorithm, the chosen demonstration is the one that allows the updated prompts to achieve the highest fairness score. This strategy balances the quality of the search with the worst-case time complexity. By accepting an increased worst-case time complexity of $O(N^2)$, the search quality is significantly enhanced. It is important to note that the G-fair-Prompting algorithm operates from a local to global perspective as shown by Algorithm 2. During the initial stages, the bias of individual samples is taken into account, while the later stages focus on reducing global predictive bias. Specifically, at each step, we insert a new demonstration $\Gamma(x_i, y_i)$ from the remaining demonstration set $S'$ (ensuring demonstrations are not repeated) at the beginning of the current context $\rho$ and select the demonstration that maximizes the fairness improvement. Formally, at step 9 in Algorithm 2, the inserted demonstration should satisfy the following criterion:

$$\arg \max_{x_i \in S'} \text{fair}(\Gamma(x_i, y_i) \oplus \rho) \quad \text{s.t. } \text{fair}(\Gamma(x_i, y_i) \oplus \rho) > \text{fair}(\rho). \tag{3}$$

---

**Algorithm 1** T-fair-Prompting

1: **Given:** training set $S = \{(x_i, y_i)\}_i^N$, pretrained LLM $\mathcal{M}$, transformation template $\Gamma(\cdot)$, and context-free input $\eta$
2: Initial prompt $\rho$
3: **for** $(x_i, y_i)$ in $S$ **do**
4:     Inference $\hat{P} \leftarrow \{\hat{p}(y|\Gamma(x_i, y_i) \oplus \eta)|y \in \mathcal{Y}\}$ via $\mathcal{M}$
5:     Calculate the $\text{fair}(\Gamma(x_i, y_i))$ according to Eq. 2
6: **end for**
7: Sort $\text{fair}_{i=1,\cdots,N}(\Gamma(x_i, y_i))$ in descending order
8: **for** $d$ in $1, \cdots, k$ **do**
9:     *Insert* the most $d$ fair demonstration at the head of $\rho$
10: **end for**
11: **return** $\rho$

**Algorithm 2** G-fair-Prompting

1: **Given:** training set $S = \{(x_i, y_i)\}_i^N$, pretrained LLM $\mathcal{M}$, transformation template $\Gamma(\cdot)$, and context-free input $\eta$
2: Initial prompt $\rho$
3: **while** $S$ *is not null* **do**
4:     **for** $(x_i, y_i)$ in $S$ **do**
5:         $\rho_{\text{tmp}} \leftarrow \Gamma(x_i, y_i) \oplus \rho$
6:         Inference $\hat{P} \leftarrow \{\hat{p}(y|\rho_{\text{tmp}} \oplus \eta)|y \in \mathcal{Y}\}$ via $\mathcal{M}$
7:         Calculate the $\text{fair}(\rho_{\text{tmp}})$ according to Eq. 2
8:     **end for**
9:     *Insert* the demonstration that can improve fairness best and *remove* it from $S$
10:     *Stop* searching when fairness can't be improved
11: **end while**
12: **return** $\rho$

---

## 5 Experiments

### 5.1 Experimental Setup

**Models.** There are a large number of available LLMs (Appendix A.2) including open-source models and black-box cloud API. Recently, Meta has released their powerful pretrained LLMs, LLaMA. LLaMA models with 13B parameters can achieve comparable performance in contrast to BLOOM and GPT-3 with much larger model size. In this paper, we evaluate the effectiveness of our method on BLOOM (176B) and LLaMA models of different sizes. We have opted to employ LLaMA (65B) as a substitute for GPT-3 in our experiments, since oepnai strictly restricts the API access to certain areas.

**Datasets.** We conducted experiments on various text classification datasets [21], namely SST-2, AGNews, CoLA, TREC, and RTE. Furthermore, the maximum input length of LLaMA is 512, and the sentences in RTE are too long for LLaMA. The task descriptions and statistics are available in Table 6 in Appendix.

### 5.2 Results

We conducted experiments on different settings and reported the results of five runs. We compared our method with the diversity-guided searching strategy proposed by Zhang et al.[12] (Global view) and the similarity-guided searching strategy proposed by Liu et al.[15] (Local view). Note that methods

Table 1: Accuracy for different prompting strategies (averaged on $5_{(0,\cdots,4)}$ different seeds, where Top-$k$ and Greedy indicate T-fair-Prompting with $k$ demonstrations and G-fair-Prompting respectively).

| Model | Dataset | Random | Diversity | Similarity | Ours Top-2 | Top-4 | Greedy |
|---|---|---|---|---|---|---|---|
| BLOOM (176B) | SST2 | $92.7_{2.3}$ | $\mathbf{95.0_{0.9}}$ | $94.0_{0.9}$ | $94.6_{0.5}$ | $93.8_{2.1}$ | $91.2_{4.0}$ |
| | AGNews | $73.9_{5.9}$ | $70.2_{10.1}$ | $74.8_{3.8}$ | $75.4_{2.2}$ | $74.8_{2.3}$ | $\mathbf{79.6_{1.4}}$ |
| | TREC | $47.9_{14.6}$ | $46.0_{8.7}$ | $31.4_{3.1}$ | $55.4_{13.3}$ | $39.2_{19.3}$ | $\mathbf{66.8_{2.5}}$ |
| | RTE | $62.4_{4.2}$ | $\mathbf{69.2_{1.9}}$ | $67.2_{3.5}$ | $55.6_{1.0}$ | $57.6_{1.9}$ | $63.0_{2.1}$ |
| | CoLA | $68.4_{4.8}$ | $\mathbf{71.0_{3.7}}$ | $69.8_{2.5}$ | $66.4_{8.6}$ | $66.8_{3.7}$ | $68.2_{6.2}$ |
| LLaMA (33B) | SST2 | $82.5_{11.8}$ | $\mathbf{90.0_{2.7}}$ | $72.8_{4.4}$ | $82.0_{11.1}$ | $80.0_{12.2}$ | $85.6_{8.2}$ |
| | AGNews | $75.2_{5.0}$ | $75.0_{5.1}$ | $75.0_{2.4}$ | $73.2_{3.9}$ | $69.8_{4.4}$ | $\mathbf{76.4_{4.6}}$ |
| | TREC | $68.1_{11.1}$ | $68.2_{4.7}$ | $60.6_{3.4}$ | $71.4_{11.1}$ | $57.8_{17.3}$ | $\mathbf{80.2_{5.3}}$ |
| | CoLA | $66.9_{11.0}$ | $68.8_{6.8}$ | $72.8_{2.0}$ | $63.8_{13.3}$ | $69.8_{3.9}$ | $\mathbf{70.6_{4.2}}$ |
| LLaMA (65B) | SST2 | $90.0_{7.7}$ | $90.8_{9.0}$ | $87.4_{3.1}$ | $88.2_{8.6}$ | $\mathbf{95.8_{1.5}}$ | $87.8_{9.0}$ |
| | AGNews | $76.8_{5.0}$ | $\mathbf{78.2_{3.1}}$ | $78.2_{1.8}$ | $77.0_{3.4}$ | $76.2_{4.9}$ | $76.0_{4.0}$ |
| | TREC | $63.6_{14.2}$ | $65.2_{10.9}$ | $64.0_{5.5}$ | $65.8_{13.0}$ | $57.4_{19.9}$ | $\mathbf{74.0_{12.2}}$ |
| | CoLA | $66.2_{9.8}$ | $62.6_{8.6}$ | $59.2_{14.0}$ | $67.6_{11.7}$ | $62.6_{6.5}$ | $\mathbf{72.0_{4.5}}$ |

based on local view are time-consuming since they require searching different demonstrations for every test example. Table 1 shows the performance of the different strategies, where "Random" indicates the average accuracy for enumerating all situations, "Diversity" and "Similarity" indicate demonstrations are selected according to diversity and similarity (details please refer to Appendix A.6), respectively. For each dataset, we set the size of the training set to 4. "Diversity" and "Similarity" select 4 from 16 demonstrations, as they need more candidates. The baseline is expensive to compute since enumerating all candidates for 4 demonstrations in RTE on BLOOM will take more than 120 NVIDIA A100 GPU hours. We enumerate all candidates for the training set with 4 demonstrations on different models, as shown in Fig. 2. The results on models whose parameters less than 13B are shown in Table 4 (i.e., GPT2-XL (1.5B), LLaMA (7B), and LLaMA (13B)).

• **G-fair-Prompting can reach a close approximation of enumeration.** To evaluate whether the G-fair-Prompting (Greedy) method can approximate the best performance of enumerating all candidates, we marked the performance of G-fair-Prompting with a green star (representing the closest value to averaged accuracy of G-fair-Prompting on the line). We found that G-fair-Prompting can achieve a very close approximation to enumeration. As shown in Fig. 2, most prompts searched by G-fair-Prompting achieved a top 20% ranking, and on BLOOM (176B), G-fair-Prompting almost found the most fair prompt.

• **G-fair-Prompting outperforms T-fair-Prompting.** As shown in Table 1, although T-fair-Prompting achieves better performance compared with random selection, G-fair-Prompting consistently outperforms T-fair-Prompting. Furthermore, Top-2 significantly outperforms Top-4 in most cases (over 5%), indicating that the number of demonstrations selected is crucial. Overall, the results demonstrate that G-fair-Prompting achieves satisfactory performance with only a slight additional cost.

• **Compared with SOTA methods.** We compared our methods with several State-of-the-Art (SOTA) methods, including diversity-guided and similarity-guided techniques. We observed that our **greedy** approach outperforms most of these SOTA methods in most situations, and the improvements of over 10% are observed on dataset TREC. The similarity-guided method, on the other hand, achieved the best performance on the topic classification task (AGNews). This is because it searches for a unique prompt for every different test example based on the dis-

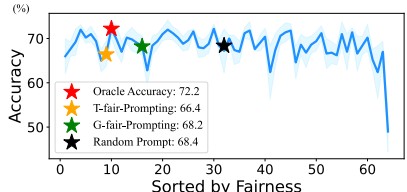

Figure 4: BLOOM is not sensitive to CoLA.

tance between the embeddings of the training samples and the test example. This strategy selects demonstrations with labels that are the same as the test samples, and Language Models (LLMs) tend to predict biased predictions toward the labels that always appear in the context. However, the similarity-guided method may prove inadequate when applied to other tasks. Specifically, the similarity-guided strategy exhibits lower performance compared to random selection in QC and acceptability tasks. Furthermore, the G-fair-Prompting approach may occasionally falter when the model's sensitivity to the task is not immediately evident, as observed in the acceptability task on BLOOM (depicted in Fig. 4). Note that the training set size of compared methods is $4\times$ larger than ours.

• **Comparison with Calibration Method.** Post-calibration [18], can enhance the accuracy of a given prompt in most cases. However, when the selected prompt is of poor quality, the performance may remain inadequate even after calibration. We compared the performance of G-fair-Prompting with random selection with calibration (averaged on all candidates), and found that G-fair-Prompting can outperform random selection with calibration in most situations. For example, on the topic classification task, G-fair-Prompting achieves the best performance on most models. Moreover, we find that post calibration can harm the performance of the model and it occurs significantly times, so it is worthwhile to reconsider the influence of manipulating the model's probability directly.

Table 2: Accuracy comparison after post calibration.

| Dataset | Method | BLOOM (176B) | | LLaMA (33B) | | LLaMA (65B) | |
|---|---|---|---|---|---|---|---|
| | | Average | Worst | Average | Worst | Average | Worst |
| TREC | Random (cal) | $66.8_{9.0}$ | 57.2 | $69.2_{6.2}$ | 59.4 | $\mathbf{74.6_{9.7}}$ | $\mathbf{66.2}$ |
| | Ours | $66.8_{2.5}$ | 64.0 | $\mathbf{80.2_{5.3}}$ | $\mathbf{75.0}$ | $74.0_{12.2}$ | 50.0 |
| | Ours (cal) | $\mathbf{77.0_{1.1}}$ | $\mathbf{75.0}$ | $76.6_{5.1}$ | 70.0 | $72.8_{12.6}$ | 48.0 |
| AGNews | Random (cal) | $73.0_{6.6}$ | 61.8 | $71.9_{5.0}$ | 64.0 | $78.2_{4.7}$ | $\mathbf{71.6}$ |
| | Ours | $\mathbf{79.6_{1.4}}$ | $\mathbf{77.0}$ | $\mathbf{76.4_{4.6}}$ | $\mathbf{69.0}$ | $76.0_{4.0}$ | 71.0 |
| | Ours (cal) | $77.4_{1.4}$ | 76.0 | $76.0_{4.4}$ | 68.0 | $76.4_{3.6}$ | 70.0 |
| CoLA | Random (cal) | $\mathbf{68.5_{5.5}}$ | $\mathbf{61.2}$ | $67.8_{5.1}$ | 63.6 | $54.0_{12.4}$ | 42.4 |
| | Ours | $68.2_{6.2}$ | 57.0 | $\mathbf{70.6_{4.2}}$ | 64.0 | $72.0_{4.5}$ | $\mathbf{66.0}$ |
| | Ours (cal) | $68.0_{5.2}$ | 58.0 | $70.4_{3.8}$ | $\mathbf{65.0}$ | $72.0_{4.5}$ | $\mathbf{66.0}$ |

Post calibration [18] can improve the accuracy of a certain prompt (in most cases), but when the selected prompt is very poor, the performance is still very poor even after calibration. We conducted experiments (Table 2) to compare the performance of G-fair-Prompting and random selection with calibration ("Average" and "Worst" indicate averaged accuracy and worst performance on all permutations of training examples), and observed that G-fair-Prompting outperforms random selection with calibration in most case. For instance, on the CoLA, G-fair-Prompting exhibited superior performance on most models. Additionally, we find that post-calibration could negatively affect the model's performance in many scenarios while it sometimes can improve the performance significantly even on selected prompts, for example, an improvement by $10\%$ is observed on BLOOM-TREC. For more detailed discussions, please refer to Appendix A.5. Hence, it is crucial to reconsider the impact of directly manipulating the model's probability.

## 6 Conclusion

In this paper, we revisit the sensitivity of large language model across prompts, and analyse the issue from a predictive bias perspective. Accordingly, we employ a "content-free" strategy as a metric termed as fairness to evaluate the predictive bias of a fixed prompt and show that model's performance is highly consistency with fairness. Then, we propose two strategy to search the most fair prompt in the original space. We conduct extensive experiments on current famous LLMs, and validate the effectiveness of the proposed strategy. Moreover, in addition to fairness adopted in this paper, there would be more metrics for prompt searching in the future for different scenarios.

## Acknowledgments

This work is supported by the National Natural Science Foundation of China (Grant No. 61976151 and 61925602), and Fu is supported by A*STAR Central Research Fund. The project was conducted during the internship in AI Lab, Tencent.

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

# A  Appendix

## A.1  Pretrained Large Language Models

Neural autoregressive language model (LMs) are designed for next token prediction to predict the probability distribution over the next token after a sequence of tokens input, and pre-trained LMs show their superior performance since they are trained on various programming languages and a large-scale curated dataset. Training large natural LMs are very expansive and time-consuming process since they always have billions of parameters, which limits the development of LMs. Fortunately, many pre-trained LMs are open access or limited access, which promotes researchers to pool their time and makes the resources to collectively achieve a higher impact. EleutherAI makes the GPT-J [22] and GPT-Neox [23] public available on Hugging Face. GPT-3 [1] is limited access in OpenAI which can be used by researchers for a fee, and another large open-science open-access multilingual language model named Bloom [2] is provided by BigScience.

## A.2  Open Access Models

Table 3: Pretrained language models

| Model | Params | Provider | Access |
|---|---|---|---|
| GPT-2 | 124 M | Hugging Face | OPEN |
| GPT-Medium | 335 M | Hugging Face | OPEN |
| GPT2-Large | 774 M | Hugging Face | OPEN |
| GPT-XL | 1.5 B | Hugging Face | OPEN |
| GPT-3 (ada) | 350 M | OPENAI | LIMITED |
| GPT-3 (babbage) | 1.3 B | OPENAI | LIMITED |
| GPT-3 (curie) | 6.7 B | OPENAI | LIMITED |
| GPT-3 (davinci) | 175 B | OPENAI | LIMITED |
| GPT-J | 6 B | EleutherAI | OPEN |
| GPT-NeoX | 20 B | EleutherAI | OPEN |
| Bloom | 176 B | BigScience | OPEN |
| LLaMA | 7 B | Meta | OPEN |
|  | 13 B | Meta | OPEN |
|  | 33 B | Meta | OPEN |
|  | 65 B | Meta | OPEN |

## A.3  Additional Figures on Different Settings

In additional to the Fig. 2, we shows the performance on different models for enumerating all candidates, note that the shadow indicates the half value of standard deviation for clear presentation since the variance is very high for LLMs.

## A.4  Accuracy Varies with demonstrations

**Accuracy Varies with Example Amount**   Demonstrations play an important role in imparting task-related information to language models through in-context learning. Then, the question arises - does a larger number of demonstrations necessarily equate to better performance? To answer this question, we evaluated performance in terms of accuracy by gradually increasing the number of demonstrations. We set $\rho = \Gamma(x_1, y_1) \oplus \cdots \oplus \Gamma(x_k, y_k)$, where $k = 1, \cdots, n$, and demonstrations are erased with $k$ decreasing from $n$ to $1$. Intuitively, accuracy would vary highly across different numbers of demonstrations, and the phenomenon is observed in Fig. 6a. To our surprise, however, erasing some demonstrations can result in a better performance. Removing some demonstrations can perform better and sometimes GPT-3 achieves best accuracy when there is only a few demonstrations remaining. This highlights the importance of considering the appropriate number of demonstrations.

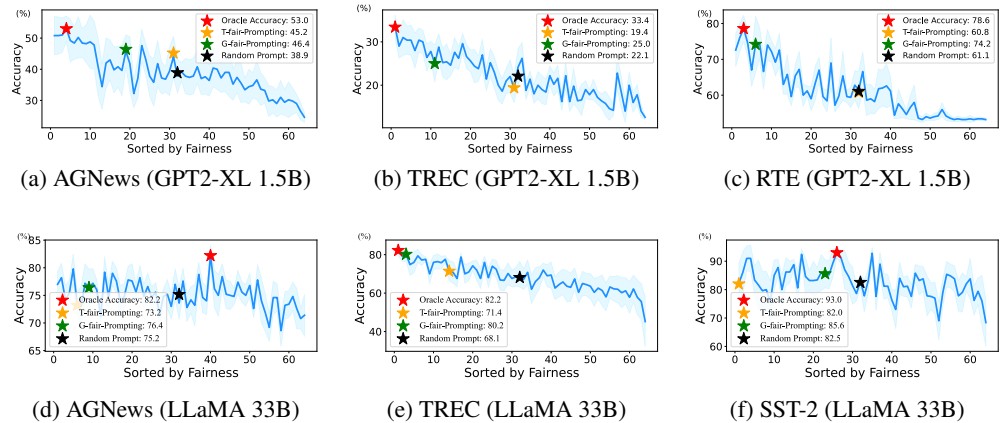

(a) AGNews (GPT2-XL 1.5B)  (b) TREC (GPT2-XL 1.5B)  (c) RTE (GPT2-XL 1.5B)

(d) AGNews (LLaMA 33B)  (e) TREC (LLaMA 33B)  (f) SST-2 (LLaMA 33B)

Figure 5: Accuracy is highly consistency with fairness and greedy search can find a good prompt, where "Random" and "Oracle" indicates the average accuracy of all prompts and the upper-bound performance according to fairness.

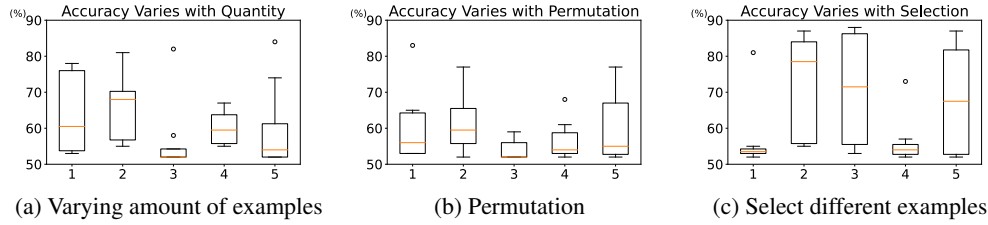

(a) Varying amount of examples  (b) Permutation  (c) Select different examples

Figure 6: ICL suffers from high instability due to variations in example amount, example order, and example selection.

**Example Order** The performance of a model is sensitive to the order of the demonstrations, as has been discussed in [4]. Even when the demonstrations are the same, different permutations of the demonstrations can result in vastly different outcomes. As there are $n!$ possible permutations, we introducing a strategy of permuting the demonstrations by circularly shifting the index of the demonstrations. The demonstration can be represented as $\rho = \Gamma(x_{k+1}, y_{k+1}) \oplus \cdots \oplus \Gamma(x_n, y_n) \oplus \Gamma(x_1, y_1) \oplus \cdots \oplus \Gamma(x_k, y_k)$. As shown in Fig. 6b, the accuracy varies highly with permutation which consistent with the observations in [4].

**Example Selection** In this paper, we find which demonstrations are selected is influence the model extremely. This scenario can be described as selecting $k$ demonstrations in $n$ training samples. In Fig. 6c, we only select one example for demonstration to ablate the impact of demonstrations order, and the accuracy also varies highly with different example selected. In this work, we only detail evaluate the proposed probing method on the erasing demonstrations and permutation, although our method improves by $20\%$ in the setting of example selection on SST-2 (GPT2-XL), because selecting $k$ demonstrations on a set with $n$ training samples can't be regarded as $k-$shot learning in the strict sense.

### A.5 Relationship between with- and without-calibration

• **G-fair-Prompting without post-calibration outperforms random demonstrations after post-calibration.** Based on Table 1, it is apparent that G-fair-Prompting outperforms random selection prior to post-calibration. This leads to a natural question: do prompts with better performance before calibration also indicate better performance after calibration proposed by Zhao et al. [18]? To investigate the relationship between performance with- and without-calibration, we calculated the Pearson correlation coefficient between the accuracy with- and without-calibration $Pearson(acc_{w/o}, acc_{with})$. A positive coefficient value suggests that a prompt with high accuracy before calibration has a

Table 4: Accuracy for different prompting strategies (averaged on $5_{0,\dots,4}$ different seeds).

| Model | Dataset | Random | Diversity | Similarity | Ours Top-2 | Ours Top-4 | Ours Greedy |
|---|---|---|---|---|---|---|---|
| GPT2-XL (1.5B) | SST-2 | $61.1_{6.1}$ | – | – | $60.8_{11.4}$ | $65.8_{8.7}$ | $74.2_{12.0}$ |
| | AGNews | $38.9_{11.4}$ | – | – | $45.2_{12.5}$ | $37.2_{11.2}$ | $46.4_{11.9}$ |
| | TREC | $22.1_{5.7}$ | – | – | $19.4_{8.9}$ | $28.2_{9.2}$ | $25.0_{7.4}$ |
| | RTE | $53.2_{6.9}$ | – | – | $54.0_{7.5}$ | $53.6_{5.9}$ | $56.4_{2.2}$ |
| LLaMA (7B) | AGNews | $64.5_{10.0}$ | $66.4_{9.1}$ | – | $66.0_{11.7}$ | $69.2_{5.5}$ | $63.8_{5.7}$ |
| | TREC | $49.5_{10.4}$ | $51.4_{9.6}$ | – | $48.4_{10.5}$ | $38.6_{15.2}$ | $61.3_{4.8}$ |
| | CoLA | $60.4_{10.6}$ | $63.8_{8.7}$ | – | $58.2_{7.8}$ | $61.6_{6.5}$ | $36.4_{3.6}$ |
| LLaMA (13B) | AGNews | $72.2_{7.7}$ | $78.4_{3.5}$ | – | $73.6_{9.0}$ | $74.2_{4.3}$ | $75.2_{2.8}$ |
| | TREC | $46.4_{16.5}$ | $48.0_{16.0}$ | – | $51.0_{16.6}$ | $39.2_{23.3}$ | $61.4_{12.1}$ |
| | CoLA | $67.7_{2.9}$ | $67.2_{2.4}$ | – | $67.0_{2.0}$ | $67.2_{1.6}$ | $67.0_{2.0}$ |

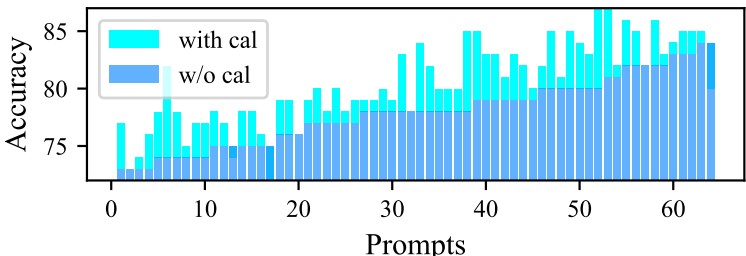

Figure 7: Illustration of accuracy relationship between with- and without calibration when $Pearson(\cdot)$ is positive.

higher likelihood of achieving higher accuracy after calibration than other prompts. We take the topic classification task on LLaMA(65B) for illustration to show the relationship between with- and without calibration when $Pearson$ is positive in Fig.7. Table 5 presents the Pearson correlation coefficient on accuracy of permutation and G-fair-Prompting after calibration. The majority of Pearson correlation coefficients were found to be positive, indicating that prompts with better performance before calibration have more potential to perform well after calibration. Furthermore, our results on the LLaMA family reveal that the larger the model, the stronger the correlation between performance with- and without-calibration. For instance, the value of the Pearson correlation coefficient increases from 0 to 0.7 as the model size increases.

**Theorem A.1.** *Suppose the performance of the model under certain prompts with- and without-calibration is positively correlated, i.e., $Pearson(acc_{w/o}, acc_{with}) > 0$, if we can assure $\mathbb{E}(acc_{w/o}^{Selected}) > \mathbb{E}(acc_{w/o}^{Random})$, then we have $\mathbb{E}(acc_{with}^{Selected}) > \mathbb{E}(acc_{with}^{Random})$.*

Table 5: Pearson's r between the with- and without-calibration.

| Dataset | BLOOM 176B | LLaMA 7B | LLaMA 13B | LLaMA 33B | LLaMA 65B |
|---|---|---|---|---|---|
| TREC | 0.1274 | 0.1551 | 0.2959 | 0.3090 | 0.5151 |
| AGNews | 0.3875 | $-0.0471$ | 0.3044 | 0.6953 | 0.7100 |
| CoLA | 0.4050 | 0.3592 | 0.5193 | 0.3611 | 0.8012 |

As analysed in Theorem A.1, if we can find a prompt with high accuracy before calibration, we have a higher likelihood of achieving higher accuracy after calibration than random selection. Our approach consistently identifies an appropriate prompt, as evidenced by the results in Table 1. Moreover, the

performance of the model exhibits a positive correlation with and without calibration under certain prompts, as illustrated in Table 5. Therefore, our method is more likely to enhance calibration performance.

## A.6 Diversity and Similarity

**Diversity** indicates the demonstrations are selected according to the diversity of the embeddings of all samples [12]. Specifically, we select the 4 most diverse samples as demonstrations by k-means clustering on training set consists of 16 samples.

**Similarity** indicates the demonstrations are selected according to the similarity of the embeddings of all samples [15]. Specifically, we select the 4 most similar samples as demonstrations according to Euclidean distance from the training set consists of 16 samples. Note that demonstrations for different test samples are different for best performance. A larger training set may result in a better performance, but we set the size of training set as 16 for a fair comparison.

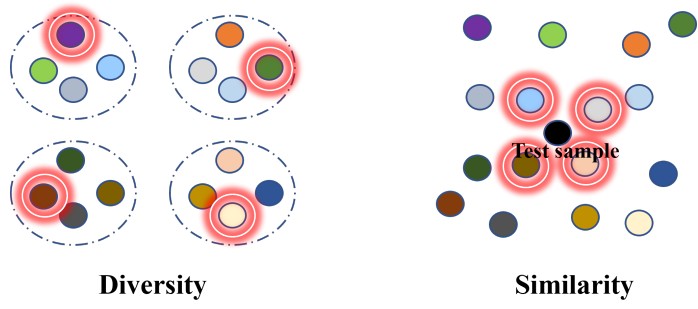

**Diversity**          **Similarity**

Figure 8: Illustration for Diversity and Similarity.

## A.7 Details for experiments

In Fig. 1, we show the high instability due to high variations in demonstrations selection and order for both with- and without-calibration [18]. Specifically, for left two figures, we sample 4 different demonstrations randomly from dataset AGNews, and estimate the influence of demonstration selection. On the other hand, the right two figures show the instable performance due to permutation when the demonstrations are fixed. Specifically, we randomly sample 4 demonstrations and estimate the performance with all possible orders.

## A.8 Complexity of different strategies

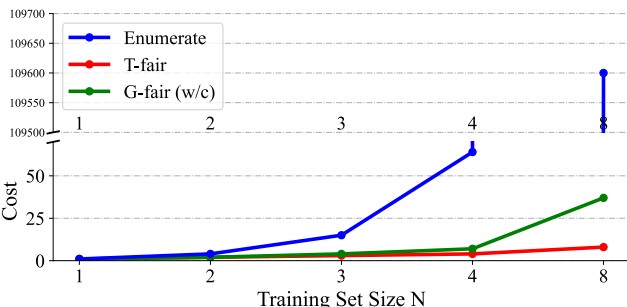

Figure 9: Computational cost. T-fair and G-fair indicate T-fair-Prompting and G-fair-Prompting respectively, and "w/c" indicates the worst case.

### A.9 Limitations.

While the proposed strategy does not require modification of the original inference process of the Language Model (LM), it still necessitates the logits or probabilities of the next token. As a result, it may be necessary to approximate the probabilities in some completely black-box LM services, such as GPT-4.

# B Clarifications After Rebuttal

*Q: Computational cost of using G-fair Prompting becomes non-trivial as N goes larger.*

(1) We acknowledge the concern regarding the computational cost of G-fair Prompting as N increases. However, we would like to emphasize that the worst-case computation cost of $1/2 * N^2$ is not a common scenario. In practice, only a few demonstrations are required. For example, in GPT-4 Technical Report [24], the number of demonstrations is fixed at 5. As a result, the overall computation cost remains acceptable, especially when compared to other existing methods [15] among the typical methods detailed in the related work.

(2) Moreover, our method does not require re-selection of demonstrations for different test samples. Consequently, the inference cost for selecting demonstrations is minimized, particularly when dealing with a larger number of test samples.

(3) In the worst-case scenario, even with an infinitely large training set, our model can efficiently perform Greedy search on a subset, consistently outperforming random demonstration. The additional computational cost depends on the size of the search space that users want to explore.

(4) In contrary, the naive strategy, i.e., enumerating all possible situations, has an exponential time complexity of O(n!). The smaller training set used in our experiments is due to considering the computational cost of the comparative methods in the report, rather than any limitations of the proposed method.

*Q: One can think of even better algorithm that can show benchmark performance that provides upper bound against G-Prompting.*

(1) The proposed simple version T-fair-prompting serves as a time-efficient alternative of G-fair-prompting, while still maintaining competitive performance.

(2) Compared with T-fair-prompting, G-fair-prompting offers the advantage of automatically determining the number of demonstrations, providing a more natural and flexible approach.

(3) Therefore, a tradeoff between performance and time cost can be achieved by transitioning between T-fair-prompting and G-fair-prompting. Specifically, we can select the demonstrations order as same as T-fair-prompting and then decide the number of demonstrations by the strategy of G-fair-prompting. The computation cost upper bound of this strategy will be the lower bound of G-fair-prompting while its fairness lower bound will be the upper bound of T-fair-prompting.

*Q: Under the proposed problem formulation, there will be a natural extension to test both different choices of few-shot examples and all possible ordering of them.*

(1) As shown in Fig.2, we have tested different choices of few-shot examples (e.g., [x1, x2]/[x3, x4]) and all possible orderings of them (e.g., [x1, x2, x4]/[x2, x4, x4]). In total, 64 different possible situations were considered when the training set size was 4.

(2) Notably, G-fair-Prompting performs remarkably close to the best performance obtained through enumeration, highlighting the efficacy of our proposed method.

*Q: How to choose the semantic-free token without any guidance?*

(1) Following previous work [18], three different semantic-free prompts are selected, i.e., ['N/A', '', ' ']. The fairness is then calculated as the average of them.

(2) These tokens were chosen based on their guaranteed performance in previous work [18]. We will provide further clarification on this selection in the final version of our paper.

*Q: Would be the proposed method helpful for long-form QA?*

While the current version of our method is designed for classification tasks, we acknowledge the potential for its extension to long-form QA scenarios.

(1) The limitation preventing direct deployment in generation tasks lies in the absence of an appropriate method to measure the fairness of sentence-level generation.

(2) Similarly, for measuring the calibration of large language models, such as ECE, it can only be calculated for classification tasks [24].

(3) However, we believe that once the community develops techniques to measure the predictive bias of sentences, our proposed strategies can be naturally extended to long sentence generation tasks. This presents an exciting avenue for future research.

*Q: What does "total cost" in table "Dataset descriptions" refer to?*

Table 6: Dataset descriptions.

| Corpus | Task | Classes | Domain | Total Cost[1] |
|--------|------|---------|--------|-----------|
| SST-2 | sentiment | 2 | movie reviews | over 60 GPU hours |
| TREC | QA/QC | 6 | open domain | over 220 GPU hours |
| AGNews | topic | 4 | news | over 250 GPU hours |
| CoLA | acceptability | 2 | misc. | over 160 GPU hours |
| RTE[2] | NLI | 2 | news, Wikipedia | over 110 GPU hours |

[1] Total Cost=Hours×GPUs. Hardware: BLOOM=A100, LLaMA=V100.
[2] Not applicable to LLaMA because of the maximum prompt token limit.

(1) "Total cost" indicates the overall computational cost on each dataset including baselines and the proposed methods.

(2) We report the total computational cost on each dataset to provide a more comprehensive description of the datasets used in this paper. We will clarify this in the final version.

*Q: The diversity baseline seems similarly competitive to the proposed approach.*

Regarding performance, we would like to clarify the following points:

(1) Our method consistently outperforms the diversity baseline across various scales of the LLaMA family datasets.

(2) The similarity in performance between the diversity method and our method on the Bloom model is attributed to the Bloom model's robustness to the transformations of the demonstration. Even random demonstrations can achieve competitive performance with the Bloom model.

(3) It is essential to note that the baselines, including the diversity method, are the latest methods proposed shortly before submitting this paper. They have demonstrated robust performance on multiple tasks.

*Q: Could the authors compare using a strategy such as the proposed one with a large language model vs. fine-tuning with a smaller model?*

** Note that although the former strategy requires iterating over the entire dataset, the latter strategy only requires performing inference. It does not incur any space cost for storing gradients nor time cost for updating gradients, making this process highly efficient.

We would like to clarify the advantages of utilizing a large language model and comparing it with fine-tuning a smaller model:

(1) Generalization ability of large models: Large language models (LLMs) accumulate significant prior knowledge during pre-training, resulting in better generalization abilities compared to small models fine-tuned for specific tasks. Even if the performance of a large model is not significantly better than that of a dedicated small model on a specific task, the robustness of the large model in open environments, such as out-of-distribution (OOD) scenarios, is significantly superior to that of a dedicated small model [24].

(2) Multiple tasks: A single large model can perform multiple tasks simultaneously. Compared to fine-tuning a small model for each task, a large model can seamlessly switch between various tasks and effortlessly incorporate new tasks.

(3) Time efficiency: Many open-source large models, such as LLaMA, do not require modifications when encountering new tasks. Even equipped by the methods including baselines in this paper, only a few additional inferences are necessary, without any updates to the model or backpropagation of gradients.

*Q: Explanation about the text in lines 293-296 and 300.*

(1) Line 293-296 Explanation:

The sentence mentioned in lines 293-296 analyzes why the similarity-guided method achieved the best performance on the topic classification task (AGNews). This method selects unique prompts for different test examples based on the distance metric. Consequently, it chooses demonstrations with labels that are the same as the test samples. Large Language Models (LLMs) tend to predict biased outcomes towards the labels that frequently appear in the context of demonstrations . As a result, the similarity-guided method excels on the topic classification task but may perform poorly on other tasks, such as natural language inference.

(2) Line 300 Explanation:

The Bloom model exhibits high robustness to various demonstrations, even random ones, leading to similar performance results between our method and the compared method. It can be observed that the Bloom model's performance remains competitive regardless of the selected demonstrations.

*Q: Discussion about limitations.*

Due to space limitations, our discussion on limitations is included in Appendix (A.9 Limitations). While the proposed strategy does not require modification of the original inference process of the Language Model (LM), it still necessitates the logits or probabilities of the next token. As a result, it may be necessary to approximate the probabilities in some completely black-box LM services, such as GPT-4.

*Q: Figure 2 shows a non-monotonic trend between the bias and predictive quality.*

According to Figure 2, the relationship between predictive bias and predictive quality is not strictly monotonic. Instead, the overall trend shows a significant negative correlation. To avoid any misunderstandings, we will use clearer descriptions in the final version of the paper.

*Q: The term fairness is slightly misleading.*

The fairness defined in our work is defined from the concept of predictive bias, which may not align precisely with the fairness, such as social bias, that is commonly referred to. It is important to note the distinction between fairness in the context of predictive bias and fairness in the context of social bias. Moreover, we highlight that the proposed metric can be extended to analyze social bias, as social bias can also lead to predictive bias [24].

*Q: Contribution and difference from [18].*

Our paper shares a similar metric with calibration-before-use [18] to access the predictive bias of a given prompt. However, the differences are significant and listed as follows:

(1) The prior approach aims to use the metric for calibrating the output, which can still be influenced by the quality of the used prompt, as evidenced by Table 3.

(2) In contrast, our research focuses on finding a near-optimal prompt on the original space to improve the model's performance, eliminating the need for any post-adjustment to the output.

(3) Additionally, our paper introduces an empirical validation of the connection between predictive bias and final task performance, as depicted in Fig. 2, which is not explored in [24].

(4) The main contribution of our work lies in analyzing the relationship between predictive bias and performance through extensive experiments, and proposing a Greedy research strategy to solve the NP-hard problem.

(5) Our experiments have revealed that even without calibration, the prompt selected by our method outperforms randomly selected prompts, even with calibration.

(6) Moreover, we have observed that [24] can potentially harm the model's performance, which is a practical concern worth considering when manipulating the model's probability directly.

*Q: How sensitive are these strategies to the choice of hyperparameters?*

In this paper, we reduce the influence of hyperparameters by minimizing the need for manual hyperparameter tuning.

(1) In G-fair-prompting, the number of demonstrations is automatically determined, indicating that no new demonstrations are added once it is determined that adding more demonstrations does not lead to improvement.

(2) As for the threshold for fairness improvement, we fix it with 0, indicating that new demonstrations are added to the prompt if they are deemed helpful.

(3) While these two aspects could be manually adjusted to make the method more flexible. To ensure the stability of the experimental results, we avoid deliberately setting these hyperparameters.

