# OpenReview forum: "Fairness-guided Few-shot Prompting for Large Language Models"
_NeurIPS.cc/2023/Conference — NeurIPS 2023 poster_

### Official Review · Reviewer_bbKr · 2023-06-24

**Soundness:** 3 good
**Presentation:** 4 excellent
**Contribution:** 2 fair
**Rating:** 5
**Confidence:** 4

**Summary:**

The paper addresses the problem of constructing an appropriate prompt for improving the performance of in-context learning in large language models (LLMs). It introduces a metric based on predictive bias to evaluate prompt quality and empirically demonstrates that prompts with higher bias lead to unsatisfactory predictive quality. Leveraging this observation, the paper proposes a novel search strategy, combining T-fair-Prompting and G-fair-Prompting, to identify near-optimal prompts for enhancing in-context learning performance. The proposed method is extensively evaluated on various downstream tasks using state-of-the-art LLMs, such as GPT-3 and LLaMA. The results show that the method effectively improves in-context learning performance in an interpretable manner.

**Strengths:**

The paper highlights the importance of prompt quality and introduces a novel metric based on predictive bias to evaluate prompt performance. The proposed method effectively searches for near-optimal prompts by addressing bias issues, leading to improved in-context learning performance.

The paper conducts comprehensive experiments on mainstream LLMs and various downstream tasks to validate the effectiveness of the proposed method. The results consistently demonstrate performance enhancements, surpassing state-of-the-art methods in most cases.

The paper presents two search strategies, T-fair-Prompting and G-fair-Prompting, which efficiently optimize prompts by considering bias and fairness. The strategies offer a balance between search quality and computational complexity, addressing the challenges of prompt construction.

The paper is well-written and effectively conveys the motivation, contributions, methodology, and experimental setup. The algorithms and search strategies are clearly described, allowing for easy understanding and reproducibility.

**Weaknesses:**

Scalability: The paper discusses the computational complexity of the search strategies and their limitations. While the proposed strategies show promising results, the complexity may become an issue when dealing with larger datasets or models. Further exploration of scalability and efficiency would be beneficial.

Generalizability: The paper mainly focuses on LLMs and downstream text classification tasks. Extending the evaluation to other types of tasks and models would enhance the generalizability of the proposed method.


**Questions:**

How generalizable is the proposed method beyond text classification tasks? Have you explored its applicability in other types of downstream tasks, such as language generation or sequence labeling?

The search strategies, T-fair-Prompting and G-fair-Prompting, effectively optimize prompts by considering bias and fairness. However, how sensitive are these strategies to the choice of hyperparameters, such as the number of demonstrations (k) or the threshold for fairness improvement? Have you conducted sensitivity analyses to investigate their impact on performance?

**Limitations:**

None.

---

> ### Author Rebuttal · Authors · 2023-08-08
>
> **Thank you for carefully reviewing our work and valuable suggestions. We believe the following point-to-point response can address all the concerns:**
>
> ---
> ***Q1: The complexity of the search strategies.***
>
> (1) We acknowledge the concern regarding the computational cost of G-fair Prompting as N increases. However, we would like to emphasize that the worst-case computation cost of 1/2*N^2 is not a common scenario. In practice, only a few demonstrations are required. For example, in GPT-4 Technical Report [1], the number of demonstrations is fixed at 5. As a result, the overall computation cost remains acceptable, especially when compared to other existing methods [2,3,4] among the typical methods detailed in the related work.
>
> (2) Moreover, our method does not require re-selection of demonstrations for different test samples.
> Consequently, the inference cost for selecting demonstrations is minimized, particularly when dealing with a larger number of test samples.
>
> (3) In the worst-case scenario, even with an infinitely large training set, our model can efficiently perform Greedy search on a subset, consistently outperforming random demonstration. The additional computational cost depends on the size of the search space that users want to explore.
>
> (4) In contrary, the naive strategy, i.e., enumerating all possible situations, has an exponential time complexity of O(n!). The smaller training set used in our experiments is due to considering the computational cost of the comparative methods in the report, rather than any limitations of the proposed method.
>
> ---
> ***Q2: Other types of tasks.***
>
> While the current version of our method is designed for classification tasks, we acknowledge the potential for its extension to other scenarios.
>
> (1) The limitation preventing direct deployment in generation tasks lies in the absence of an appropriate method to measure the fairness of sentence-level generation.
>
> (2) Similarly, for measuring the calibration of large language models, such as ECE, it can only be calculated for classification tasks [1].
>
> (3) However, we believe that once the community develops techniques to measure the predictive bias of sentences, our proposed strategies can be naturally extended to long sentence generation tasks. This presents an exciting avenue for future research.
>
>
> ---
> ***Q3: How sensitive are these strategies to the choice of hyperparameters?***
>
> In this paper, we reduce the influence of hyperparameters by minimizing the need for manual hyperparameter tuning.
>
> (1) In G-fair-prompting, the number of demonstrations is automatically determined, indicating that no new demonstrations are added once it is determined that adding more demonstrations does not lead to improvement.
>
> (2) As for the threshold for fairness improvement, we fix it with 0, indicating that new demonstrations are added to the prompt if they are deemed helpful.
>
> (3) While these two aspects could be manually adjusted to make the method more flexible. To ensure the stability of the experimental results, we avoid deliberately setting these hyperparameters.
>
> ---
>
> [1] OpenAI. GPT-4 Technical Report. URL: https://arxiv.org/pdf/2303.08774v3.pdf
>
> [2] Yao Lu. Fantastically ordered prompts and where to find them: Overcoming few-shot prompt order sensitivity. In ACL, 2021.
>
> [3] Claudio Gentile. Fast rates in pool-based batch active learning. In ICML 2022.
>
> [4] Jiachang Liu. What makes good in-context examples for gpt-3? In ACL, 2022.

---

### Official Review · Reviewer_35hW · 2023-07-03

**Soundness:** 3 good
**Presentation:** 2 fair
**Contribution:** 3 good
**Rating:** 5
**Confidence:** 3

**Summary:**

The paper examines the sensitivity of large language models to prompts and analyzes the issue from a predictive bias standpoint. They utilize a fairness metric called a content-free strategy to evaluate the predictive bias of a fixed prompt. Methodologically, the paper proposes two strategies, T-fair-Prompting and G-fair-Prompting, to search for the fairest prompt and validates their effectiveness through extensive experiments on large-language models.

**Strengths:**

1. The paper exploits the correlation between predictive bias and overall performance and proposes two strategies based on the finding to approximately estimate the fairest demonstration combination.
2. It is intriguing to observe that fairness-guided selection, rather than diversity or similarity-guided approach, often exhibits better performance.

**Weaknesses:**

1. The predictive bias-based fairness metric, which is the most key and primary idea, is adopted directly from [18]. The way of using the fairness metric in each T-fair-Prompting and G-fair-Prompting is also straight-forward. I think that the methodological novelty of this paper is limited.
2. I can't find an ablation study regarding the proposed fairness metric. I am curious about the results and analysis when using other semantic-free prompts, apart from N/A.
3. When looking at Table 2, there are quite a few results where diversity-guided selection demonstrates higher performance. The overall performance is not particularly impressive.

**Questions:**

Please provide an in-depth response regarding the weaknesses mentioned above.

**Limitations:**

The role of few-shot prompting in in-context learning is significant. Therefore, I believe that providing a new insight is beneficial for the in-context learning community. However, I would like to request a sufficient response that adequately addresses the novelty of the proposed work.

---

> ### Author Rebuttal · Authors · 2023-08-08
>
> **Thank you for your time and valuable suggestions. Next, we will answer the questions in detail point by point to address any concerns.**
>
> ---
> ***Q1: Contribution and difference from [1].***
>
> Our paper shares a similar metric with calibration-before-use [1] to access the predictive bias of a given prompt. However, the differences are significant and listed as follows:
>
> (1) The prior approach aims to use the metric for calibrating the output, which can still be influenced by the quality of the used prompt, as evidenced by Table 3.
>
> (2) In contrast, our research focuses on finding a near-optimal prompt on the original space to improve the model's performance, eliminating the need for any post-adjustment to the output.
>
> (3) Additionally, our paper introduces an empirical validation of the connection between predictive bias and final task performance, as depicted in Fig. 2, which is not explored in [1].
>
> (4) The main contribution of our work lies in analyzing the relationship between predictive bias and performance through extensive experiments, and proposing a Greedy research strategy to solve the NP-hard problem.
>
> (5) Our experiments have revealed that even without calibration, the prompt selected by our method outperforms randomly selected prompts, even with calibration.
>
> (6) Moreover, we have observed that [1] can potentially harm the model's performance, which is a practical concern worth considering when manipulating the model's probability directly.
>
> ---
>
> ***Q2: About semantic-free prompts.***
>
> (1) Following previous work [1], three different semantic-free prompts are selected, i.e., [‘N/A’, ‘’, ‘ ’], and the fairness is calculated as the average of them.
>
> (2) We choose these tokens to represent semantic-free token as they demonstrate guaranteed performance in previous work [1].
>
> ---
> ***Q3: The overall performance is not particularly impressive.***
>
> Regarding performance, we would like to clarify the following points:
>
> (1) Our method consistently outperforms the diversity baseline across various scales of the LLaMA family datasets.
>
> (2) The similarity in performance between the diversity method and our method on the Bloom model is attributed to the Bloom model's robustness to the transformations of the demonstration. Even random demonstrations can achieve competitive performance with the Bloom model.
>
> (3) It is essential to note that the baselines, including the diversity method, are the latest methods proposed shortly before submitting this paper. They have demonstrated robust performance on multiple tasks.
>
> ---
>
> [1] Tony Z. Zhao. Calibrate before use: Improving few-shot performance of language models. In ICML, 2021.

---

> > ### Comment · Reviewer_35hW · 2023-08-20
> >
> > Thanks authors for their detailed clarifications. I have also reviewed the comments made by other reviewers and the authors' responses to them. The authors' responses addressed my concerns, and as a result, I will consider increasing my score to "borderline accept".

---

### Official Review · Reviewer_WoAP · 2023-07-05

**Soundness:** 4 excellent
**Presentation:** 3 good
**Contribution:** 3 good
**Rating:** 6
**Confidence:** 4

**Summary:**

The paper proposes two new procedures to select examples to be used for few-shot learning (in-context learning) for LLMs. The key idea for example selection is to select those examples that lead to a distribution over classes that is as close to uniform as possible . The two proposed variants (a) select the top k examples which each individually have the least ‘bias’ and (b) greedily add examples that reduce ‘bias’ the most until any additional example would increase it. The authors evaluate the proposed methods on a variety of open source LLMs on multiple tasks, where the proposed methods perform well.

**Strengths:**

In-context learning, and in particular the selection of effective demonstrations is an important area of study and thus of interest to the research community.
The paper is well written and the proposed methods are clearly presented.
The evaluation is well chosen to demonstrate the effectiveness of the proposed methods.
The proposed methods are a clever idea that is simple yet effective.

**Weaknesses:**

The proposed method is limited to multi-class classification tasks, which places a significant limit on the practical use-cases of the proposed method, i.e., it does not apply to summarization, translation, open ended generation etc.
Some of the claims in the paper are too strong and not supported. For example in the abstract in lines 9-11 the authors claim that “Then we empirically show that prompts with higher bias always lead to unsatisfactory predictive quality.” Which is clearly not true and not supported by their findings in Figure 2, which shows a non-monotonic trend.
While the writing is generally good, there are a few mistakes across the paper, e.g., line 34 to construct

I think the use of the term fairness is slightly misleading in this paper. This the term ‘fairness’ is associated with many different meanings, but mostly with social aspects, the authors could directly use entropy as that would be a more technical description of their method.

**Questions:**

I would love to see the authors address the usage of their method to other tasks that are not easily cast into a multi-class classification task.
I would also like the authors to address the use of the term ‘fairness’ and why that is necessary for the paper at hand.

**Limitations:**

The authors did not address limitations and other impact. Especially, given the term fairness in the title, I would expect the authors to at least discuss the use of the proposed method to address fairness issues and also the potential misuse to introduce unwanted biases.

---

> ### Author Rebuttal · Authors · 2023-08-08
>
> **Thank you for taking the time to provide valuable suggestions and have a positive attitude towards our work.**
>
> ---
>
> ***Q1: The proposed method is limited to multi-class classification tasks.***
>
> (1) We acknowledge the limitation that the proposed method is currently applicable only to classification tasks.
>
> (2) The primary reason behind this limitation is the lack of an appropriate method to measure the fairness of sentence-level generation in the current context.
>
> (3) Additionally, the calibration measurement, such as Expected Calibration Error (ECE), is feasible only for classification tasks [1].
>
> (4) However, we believe that as the research community develops methods to measure predictive bias in sentences, the strategies proposed in our paper can be naturally extended to long sentence generation tasks.
>
> ---
> ***Q2: Figure 2 shows a non-monotonic trend between the bias and predictive quality.***
>
> According to Figure 2, the relationship between predictive bias and predictive quality is not strictly monotonic. Instead, the overall trend shows a significant negative correlation. To avoid any misunderstandings, we will use clearer descriptions in the final version of the paper.
>
> ---
> ***Q3: The term fairness is slightly misleading.***
>
> The fairness defined in our work is defined from the concept of predictive bias, which may not align precisely with the fairness, such as social bias, that is commonly referred to.
> It is important to note the distinction between fairness in the context of predictive bias and fairness in the context of social bias.
> Moreover, we highlight that the proposed metric can be extended to analyze social bias, as social bias can also lead to predictive bias [1].
>
> ---
> [1] OpenAI. GPT-4 Technical Report. URL: https://arxiv.org/pdf/2303.08774v3.pdf

---

> > ### Comment · Reviewer_WoAP · 2023-08-18
> > **Thanks for the response**
> >
> > I would like to thank the authors for the extensive and detailed rebuttal - it has been very helpful.
> > I continue to think that this work is of interest.

---

### Official Review · Reviewer_pNzj · 2023-07-27

**Soundness:** 3 good
**Presentation:** 3 good
**Contribution:** 3 good
**Rating:** 6
**Confidence:** 4

**Summary:**

The paper shows that the accuracy of large language models on various text classification tasks for a given prompt tends to be correlated with the entropy of the output prediction when using that prompt for a context free input: in general, the more uniform the output probability distribution for a given prompt (made up of demonstrations from the dataset), the higher the accuracy of that prompting strategy for the dataset. The authors use this insight to select the demonstrations and show that greedily selecting demonstrations one-by-one can yield competitive performance to other strategies.

**Strengths:**

1. The authors point out that empirical observation that higher entropy in the context-free case is correlated with higher accuracy was first introduced in Zhao et al. 2021. However, showing this correlation holds in the context of prompting is, to my knowledge, novel.
2. The proposed prompting search strategies are simple to implement and competitive.
3. The experiments seem pretty extensive, both in datasets and models tested.

**Weaknesses:**

1. Given the time complexity of the G-fair-prompting algorithm (i.e. O(N^2) where N is the size of the training dataset), the proposed approach does not seem very practical.
2. The discussion of experimental results is vague at times. For instance, the explanation provided for why the similarity-guided method is best for AGNews (lines 293-296) seems as if it would apply to other tasks as well; a meaningful explanation would be able to differentiate between these settings. Also, it is not clear what "the model's sensitivity to the task is not immediately evident" means (line 300). In addition, practical discussion of the proposed approach relative to baselines would be helpful (e.g. when one is preferred over the other).

**Questions:**

1. What does “Total Cost” in Table 1 refer to? What are the time complexities of the G-fair-prompting approach on the models and datasets tested?
2. The diversity baseline seems similarly competitive to the proposed approach. When would the authors suggest using one over the other?
3. Could the authors compare using a strategy such as the proposed one with a large language model vs. fine-tuning with a smaller model? Especially since the former requires iterating over the entire dataset anyway, albeit differently than in fine-tuning, when is the former advantageous over the latter?
4. Could the authors explain the text in lines 293-296 and 300 (as per comment #2 in weaknesses)?

**Limitations:**

The authors do not directly address limitations, which would be worth mentioning briefly. I did not factor this absence into my review.

---

> ### Author Rebuttal · Authors · 2023-08-08
>
> ***Q1: The time complexity of G-fair-prompting.***
>
> (1) We acknowledge the concern regarding the computational cost of G-fair Prompting as N increases. However, we would like to emphasize that the worst-case computation cost of 1/2*N^2 is not a common scenario. In practice, only a few demonstrations are required. For example, in GPT-4 Technical Report [1], the number of demonstrations is fixed at 5. As a result, the overall computation cost remains acceptable, especially when compared to other existing methods [2,3,4].
>
> (2) Moreover, our method does not require re-selection of demonstrations for different test samples.
> Consequently, the inference cost for selecting demonstrations is minimized, particularly when dealing with a larger number of test samples.
>
> (3) In the worst-case scenario, even with an infinitely large training set, our model can efficiently perform Greedy search on a subset, consistently outperforming random demonstration. The additional computational cost depends on the size of the search space that users want to explore.
>
> (4) In contrary, the naive strategy, i.e., enumerating all possible situations, has an exponential time complexity of O(n!). The smaller training set used in our experiments is due to considering the computational cost of the comparative methods in the report, rather than any limitations of the proposed method.
>
> ---
>
> ***Q2: What does “total cost” in table 1 refer to?***
>
> (1) “Total cost” indicates the overall computational cost on each dataset including baselines and the proposed methods.
>
> (2) We report the total computational cost on each dataset to provide a more comprehensive description of the datasets used in this paper. We will clarify this in the final version.
>
> ---
> ***Q3: The diversity baseline seems similarly competitive to the proposed approach.***
>
> Regarding performance, we would like to clarify the following points:
>
> (1) Our method consistently outperforms the diversity baseline across various scales of the LLaMA family datasets.
>
> (2) The similarity in performance between the diversity method and our method on the Bloom model is attributed to the Bloom model's robustness to the transformations of the demonstration. Even random demonstrations can achieve competitive performance with the Bloom model.
>
> (3) It is essential to note that the baselines, including the diversity method, are the latest methods proposed shortly before submitting this paper. They have demonstrated robust performance on multiple tasks.
>
> ---
> ***Q4: Could the authors compare using a strategy such as the proposed one with a large language model vs. fine-tuning with a smaller model?***
>
> ** Note that although the former strategy requires iterating over the entire dataset, the latter strategy only requires performing inference. It does not incur any space cost for storing gradients nor time cost for updating gradients, making this process highly efficient.
>
> We would like to clarify the advantages of utilizing a large language model and comparing it with fine-tuning a smaller model:
>
> (1) Generalization ability of large models: Large language models (LLMs) accumulate significant prior knowledge during pre-training, resulting in better generalization abilities compared to small models fine-tuned for specific tasks. Even if the performance of a large model is not significantly better than that of a dedicated small model on a specific task, the robustness of the large model in open environments, such as out-of-distribution (OOD) scenarios, is significantly superior to that of a dedicated small model [1].
>
> (2) Multiple tasks: A single large model can perform multiple tasks simultaneously. Compared to fine-tuning a small model for each task, a large model can seamlessly switch between various tasks and effortlessly incorporate new tasks.
>
> (3) Time efficiency: Many open-source large models, such as LLaMA, do not require modifications when encountering new tasks. Even equipped by the methods including baselines in this paper, only a few additional inferences are necessary, without any updates to the model or backpropagation of gradients.
>
> ---
> ***Q5: Explanation about the text in lines 293-296 and 300.***
>
> (1) Line 293-296 Explanation:
>
> The sentence mentioned in lines 293-296 analyzes why the similarity-guided method achieved the best performance on the topic classification task (AGNews). This method selects unique prompts for different test examples based on the distance metric. Consequently, it chooses demonstrations with labels that are the same as the test samples. Large Language Models (LLMs) tend to predict biased outcomes towards the labels that frequently appear in the context of demonstrations .
> As a result, the similarity-guided method excels on the topic classification task but may perform poorly on other tasks, such as natural language inference.
>
> (2) Line 300 Explanation:
>
> The Bloom model exhibits high robustness to various demonstrations, even random ones, leading to similar performance results between our method and the compared method. It can be observed that the Bloom model's performance remains competitive regardless of the selected demonstrations.
>
> ---
> ***Q6: Discussion about limitations.***
>
> Due to space limitations, our discussion on limitations is included in Appendix (A.9 Limitations). While the proposed strategy does not require modification of the original inference process of the Language Model (LM), it still necessitates the logits or probabilities of the next token. As a result, it may be necessary to approximate the probabilities in some completely black-box LM services, such as GPT-4.
>
> ---
>
> [1] OpenAI. GPT-4 Technical Report.
>
> [2] Yao Lu. Fantastically ordered prompts and where to find them: Overcoming few-shot prompt order sensitivity. In ACL, 2021.
>
> [3] Claudio Gentile. Fast rates in pool-based batch active learning. In ICML 2022.
>
> [4] Jiachang Liu. What makes good in-context examples for gpt-3? In ACL, 2022.

---

> > ### Comment · Reviewer_pNzj · 2023-08-15
> > **Thanks for the response**
> >
> > I appreciate the response. I am happy to maintain my score.
> >
> > A few quick clarifying question: what does this point refer to? ** Note that although the former strategy requires iterating over the entire dataset, the latter strategy only requires performing inference. It does not incur any space cost for storing gradients nor time cost for updating gradients, making this process highly efficient.

---

> > > ### Author Response · Authors · 2023-08-15
> > > **Thanks for your response**
> > >
> > > We are pleased to provide further elaboration on this statement.
> > >
> > > While adjusting demonstrations requires iterating over the entire dataset, the network weights of LLMs remain fixed. This process does not involve updating the network weights; it only requires obtaining the corresponding output results for the given demonstrations.
> > >
> > > In other words, such methods do not necessitate memory to store the network gradients nor do they require expensive computations to update the network weights of the model.

---

### Official Review · Reviewer_Q4AG · 2023-07-29

**Soundness:** 3 good
**Presentation:** 3 good
**Contribution:** 3 good
**Rating:** 4
**Confidence:** 3

**Summary:**

Finding an (nearly) optimal prompt is crucial for successful in-context learning (ICL) of large language models (LLMs). It is known that performance of downstream tasks is sensitive to prompts themselves, few-shot examples, and their orders, often requiring large degree of non-trivial tuning. This paper presents a content-agnostic fairness metric that allows users to assess the quality of a given prompt purely in the perspective of predictive bias. Under this metric, the authors propose two methods for optimizing the prompts. Given the few-shot examples for ICL, T-fair Prompting computes the predictive bias of individual demonstrations by feeding each single shot. Then it chooses the user-specified number of examples with the lowest predictive biases. In contrast, G-fair Prompting promotes the greedy local search where it appends one additional shot that shows the best overall fairness. Users can choose one of the two proposed methods considering the tradeoff between algorithmic complexity and performance gain.

**Strengths:**

(1)	The paper provides simple yet effective approaches for prompt tuning.

(2)	Both T-fair and G-fair Prompting algorithms are natural and practical in the spirit of local individual search and greedy search.

(3)	The proposed methods achieve highly competitive accuracies comparing to the current state-of-the-art methods.

**Weaknesses:**

(1)	Computational cost of using G-fair Prompting becomes non-trivial as N goes larger even if it uses greedy approach for expanding local bias to global bias.

(2)	One can think of even better algorithm that can show benchmark performance that provides upper bound against G-Prompting.

(3)	Neither guidance for how to choose semantic-free tokens nor sensitivity analysis for various semantic-free candidates are explored.

**Questions:**

(1)	Can you provide computational cost for running T-fair and G-fair prompting over increasing N?

(2)	Under the proposed problem formulation, there will be a natural extension to test both different choices of few-shot examples and all possible ordering of them. Though such experiments would be expansive to navigate combinatorially many configurations, measuring the performance on a limited setting helps understanding the relative performance of T-fair and G-fair Prompting.

(3)	How to choose the semantic-free token without any guidance? It is less clear that such token would exist universally across different LLMs across various downstream tasks. In any case, using just one manually-reserved tokens for fairness evaluation is not easily justifiable without sensitivity analysis.

(4)	Predictive bias would be more useful for solving classification tests by generation (as if the paper tested the sentiment analysis). Would be the proposed method helpful for long-form QA?

**Limitations:**

No specific points are described or probed.

---

> ### Author Rebuttal · Authors · 2023-08-08
>
> **Thank you for the reviewers' valuable feedback. We have carefully reviewed the comments and offer the following responses to address each point:**
>
> ----
>
> ***Q1: Computational cost of using G-fair Prompting becomes non-trivial as N goes larger.***
>
>
> (1) We acknowledge the concern regarding the computational cost of G-fair Prompting as N increases. However, we would like to emphasize that the worst-case computation cost of 1/2*N^2 is not a common scenario. In practice, only a few demonstrations are required. For example, in GPT-4 Technical Report [1], the number of demonstrations is fixed at 5. As a result, the overall computation cost remains acceptable, especially when compared to other existing methods [2,3,4] among the typical methods detailed in the related work.
>
> (2) Moreover, our method does not require re-selection of demonstrations for different test samples.
> Consequently, the inference cost for selecting demonstrations is minimized, particularly when dealing with a larger number of test samples.
>
> (3) In the worst-case scenario, even with an infinitely large training set, our model can efficiently perform Greedy search on a subset, consistently outperforming random demonstration. The additional computational cost depends on the size of the search space that users want to explore.
>
> (4) In contrary, the naive strategy, i.e., enumerating all possible situations, has an exponential time complexity of O(n!). The smaller training set used in our experiments is due to considering the computational cost of the comparative methods in the report, rather than any limitations of the proposed method.
>
> ----
>
> ***Q2: One can think of even better algorithm that can show benchmark performance that provides upper bound against G-Prompting.***
>
>
> (1) The proposed simple version T-fair-prompting serves as a time-efficient alternative of G-fair-prompting, while still maintaining competitive performance.
>
> (2) Compared with T-fair-prompting, G-fair-prompting offers the advantage of automatically determining the number of demonstrations, providing a more natural and flexible approach.
>
> (3) Therefore, a tradeoff between performance and time cost can be achieved by transitioning between T-fair-prompting and G-fair-prompting. Specifically, we can select the demonstrations order as same as T-fair-prompting and then decide the number of demonstrations by the strategy of G-fair-prompting. The computation cost upper bound of this strategy will be the lower bound of G-fair-prompting while its fairness lower bound will be the upper bound of T-fair-prompting.
>
> ----
>
> ***Q3: Under the proposed problem formulation, there will be a natural extension to test both different choices of few-shot examples and all possible ordering of them.***
>
>
> (1) As shown in Fig.2, we have tested different choices of few-shot examples (e.g., [x1, x2]/[x3, x4]) and all possible orderings of them (e.g., [x1, x2, x4]/[x2, x4, x4]). In total, 64 different possible situations were considered when the training set size was 4.
>
> (2) Notably, G-fair-Prompting performs remarkably close to the best performance obtained through enumeration, highlighting the efficacy of our proposed method.
>
> ----
>
> ***Q4: How to choose the semantic-free token without any guidance?***
>
>
> (1) Following previous work [5], three different semantic-free prompts are selected, i.e., [‘N/A’, ‘’, ‘ ’]. The fairness is then calculated as the average of them.
>
> (2) These tokens were chosen based on their guaranteed performance in previous work [1]. We will provide further clarification on this selection in the final version of our paper.
>
> ----
>
> ***Q5: Would be the proposed method helpful for long-form QA?***
>
> While the current version of our method is designed for classification tasks, we acknowledge the potential for its extension to long-form QA scenarios.
>
> (1) The limitation preventing direct deployment in generation tasks lies in the absence of an appropriate method to measure the fairness of sentence-level generation.
>
> (2) Similarly, for measuring the calibration of large language models, such as ECE, it can only be calculated for classification tasks [1].
>
> (3) However, we believe that once the community develops techniques to measure the predictive bias of sentences, our proposed strategies can be naturally extended to long sentence generation tasks. This presents an exciting avenue for future research.
>
> ---
>
> [1] OpenAI. GPT-4 Technical Report. URL: https://arxiv.org/pdf/2303.08774v3.pdf
>
> [2] Yao Lu. Fantastically ordered prompts and where to find them: Overcoming few-shot prompt order sensitivity. In ACL, 2021.
>
> [3] Claudio Gentile. Fast rates in pool-based batch active learning. In ICML 2022.
>
> [4] Jiachang Liu. What makes good in-context examples for gpt-3? In ACL, 2022.
>
> [5] Tony Z. Zhao. Calibrate before use: Improving few-shot performance of language models. In ICML, 2021.

---

> ### Comment · Reviewer_Q4AG · 2023-08-18
> **Thanks for your response.**
>
> Thanks the authors for their step-by-step feedback. As some points get clarified, I have increased the soundness score. But its main usefulness toward the generation task is unclear, limiting the scope of practicality. Though I would not be against the positive evaluations, my rating will remain same.

---

> > ### Author Response · Authors · 2023-08-21
> > **Thank you for your response.**
> >
> > Thank you for your response. In regards to the concerns raised regarding the generation task, we would like to provide the following clarifications.
> >
> > (1) Currently, there is no universally accepted definition for evaluating the quality of generated text in the context of generation tasks. So some studies employ the use of "GPT4" to provide a scoring mechanism as a proxy measure[1][2][3].
> >
> > (2) The method proposed in this paper can be extended to generation tasks by utilizing proxy metrics. For instance, one can employ the KL divergence between the probability distributions generated by LLMs and a uniform distribution as a proxy measure of predictive bias.
> >
> > (3) Furthermore, the demonstration search strategy proposed in this paper is versatile and can be tailored to specific requirements by selecting different metrics as search objectives. For instance, in addition to considering the predictive bias of generated answers in a generation task, one can also incorporate the diversity of generated answers as a criterion for selecting demonstrations.
> >
> > [1] Tim Dettmers, Artidoro Pagnoni, Ari Holtzman, and Luke Zettlemoyer. Qlora: Efficientfinetuning of quantized llms. arXiv preprint arXiv:2305.14314, 2023.
> >
> > [2] Chunting Zhou, Pengfei Liu, Puxin Xu, Srini Iyer, Jiao Sun, Yuning Mao, Xuezhe Ma, Avia Efrat, Ping Yu, Lili Yu, et al. Lima: Less is more for alignment. arXiv preprint arXiv:2305.11206, 2023.
> >
> > [3] Arnav Gudibande, Eric Wallace, Charlie Snell, Xinyang Geng, Hao Liu, Pieter Abbeel, Sergey Levine, and Dawn Song. The false promise of imitating proprietary llms. arXiv preprint arXiv:2305.15717, 2023.

---

### Decision · Program_Chairs · 2023-09-21

**Decision:**

Accept (poster)

**Comment:**

This work proposes two new algorithms to select examples to be used in few-shot learning. Example selection is quite important for few-shot learning and this work shows that selecting examples that lead to a distribution over classes that is as close to uniform as possible increases classification performance. The two proposed variants are to select the top k examples that deviate the least from the uniform distribution and to greedily add examples that reduce deviation the most. Good results are shown on open source LLMs. The word 'fairness' is misleading in this paper, it would be better to use a different word such as entropy.

Strengths:
- ICL is an important area and this work adds to the understanding of how examples selection affects ICL performance.
- The work is well written and presentation is good.
- The idea that higher entropy correlates with higher accuracy for prompting appears to be novel.
- The approach is simple to implement and yields good results.

Weaknesses:
- The term 'fairness' is used in a misleading way.
- The method is only applicable to multi-class classification tasks and cannot be generalized to generative tasks.
- Some claims in the paper are too strong.
- The G-fair prompting algorithm may be impractical.
- Discussion of experimental results is a bit vague.